# Expanding the Toolbox for Bicelle-Forming Surfactant–Lipid Mixtures

**DOI:** 10.3390/molecules27217628

**Published:** 2022-11-07

**Authors:** Rita Del Giudice, Nicolò Paracini, Tomas Laursen, Clement Blanchet, Felix Roosen-Runge, Marité Cárdenas

**Affiliations:** 1Department of Biomedical Science and Biofilms Research Center for Biointerfaces, Malmo University, 205 06 Malmo, Sweden; 2Plant Biochemistry, Department of Plant and Environmental Sciences, University of Copenhagen, Thorvaldsensvej 40, DK-1871 Copenhagen, Denmark; 3European Molecular Biology Laboratory (EMBL) Hamburg Outstation, DESY, 22607 Hamburg, Germany; 4Department of Biological Sciences, Nanyang Technological University, Singapore 639798, Singapore; 5Biofisika Institute (CSIC, UPV/EHU), 48940 Leioa, Spain

**Keywords:** bicelles, SAXS, model membranes, DLS, cryo-TEM

## Abstract

Bicelles are disk-shaped models of cellular membranes used to study lipid–protein interactions, as well as for structural and functional studies on transmembrane proteins. One challenge for the incorporation of transmembrane proteins in bicelles is the limited range of detergent and lipid combinations available for the successful reconstitution of proteins in model membranes. This is important, as the function and stability of transmembrane proteins are very closely linked to the detergents used for their purification and to the lipids that the proteins are embedded in. Here, we expand the toolkit of lipid and detergent combinations that allow the formation of stable bicelles. We use a combination of dynamic light scattering, small-angle X-ray scattering and cryogenic electron microscopy to perform a systematic sample characterization, thus providing a set of conditions under which bicelles can be successfully formed.

## 1. Introduction

Bicelles are nano-sized discoidal-like particles composed of long-chain phospholipids, which form a lipid bilayer core, and detergents or short-chain phospholipids, which form the flanking rim. These structures spontaneously assemble by mixing phospholipids and a detergent at a given lipid:detergent molar ratio (q), which determines the particle size, and their properties vary depending on experimental conditions, such as lipid and detergent type and concentration, buffer ionic strength, and temperature [1,2,3]. Their high compositional tunability, along with stability against several freeze–thawing cycles, makes bicelles popular as models of cellular membranes for the study of both transmembrane protein structure and functioning and their intermolecular interactions in a native-like environment [4,5].

The size of bicelles can be tuned by altering the lipid:detergent molar ratio (q), with the size increasing at higher q values. At q > 1, bicelles become more elongated and such elongated structures make bicelles suitable for nuclear magnetic resonance (NMR) studies due to the alignment of bicelles along the magnetic field [1,6].

Bicelles composed of the zwitterionic phospholipid 1,2-dimyristoyl-*sn*-glycero-3-phosphocholine (DMPC) and the detergent 1,2-dihexanoyl-*sn*-glycero-3-phosphocholine (DHPC) constitute the most extensively studied system. Studies of their ultrastructure and morphology by small-angle X-ray scattering (SAXS), small-angle neutron scattering (SANS), dynamic light scattering (DLS), and cryogenic electron microscopy (cryo-EM) led to the classical description of bicelles as a bilayer disk composed of DMPC phospholipids surrounded by a rim composed of the short-chain DHPC surfactant [7,8,9,10,11].

Different detergents and phospholipid combinations have been used to produce bicelles. These include the anionic 1,2-dimyristoyl-*sn*-glycero-3-phospho-*sn*-glycerol (DMPG) and the zwitterionic 1,2-dipalmitoyl-*sn*-glycero-3-phosphocholine (DPPC) [12]. As for the surfactants used, bicelles have been produced for NMR studies [13,14] using 3-[(3-cholamidopropyl)dimethylammonio]-1-propanesulfonate (CHAPS) and 3-[(3-cholamidopropyl)dimethylammonio]-2-hydroxy-1-propanesulfonate (CHAPSO), surfactants routinely used in membrane protein purification, given their ability to preserve protein structural integrity. Bicelles developed using DHPC as well as fatty acids and other detergents have also been used to form solid-supported lipid bilayers [15,16].

Despite the wide range of applications for bicelles as models of biomembranes, detailed structural studies exist only for a handful of compositions [2,17]. The standard bicelle model is composed of a disk with three layers (phospholipid head/tail/head) and a rim (detergent head), but more detailed models have been produced to mimic different regions of the lipid core with varying molecular packings [2]. In this work, we applied DLS, SAXS, and cryo-EM to characterize bicelles developed using either saturated (DMPC/DMPG) or unsaturated 2-oleoyl-1-palmitoyl-sn-glyecro-3-phosphocholine (POPC) and 1-palmitoyl-2-oleoyl-sn-glycero-3-phospho-(1′-rac-glycerol) (POPG) phospholipids combined with three different detergents (DHPC, CHAPS, and sodium cholate). We identify the conditions suitable for controlling the particle size and shape of bicelles composed of the components shown in Figure 1.

## 2. Results

The aim of this study was to systematically characterize bicelles composed of different combinations of lipids and detergents to determine the conditions under which stable bicelles can be formed and potentially used for the in-solution study of transmembrane proteins. In this study three different detergents were investigated: CHAPS and sodium cholate, which were chosen as they have been widely used for the purification of membrane proteins, and DHPC, which has been extensively used to form bicelles in the past.

For each of the three detergents, bicelles were prepared with four lipid compositions: (i) saturated zwitterionic (DMPC), (ii) saturated zwitterionic/anionic (DMPC:DMPG 80:20), (iii) unsaturated zwitterionic (POPC), and (iv) unsaturated zwitterionic/anionic (POPC:POPG 80:20) (Figure 1). Each detergent–lipid combination was prepared at seven different q ratios: 0.5, 1, 1.5, 2, 2.5, 3, and 5 by resuspending lipid/detergent films in a buffer containing the detergent of choice at a concentration slightly above the critical micelle concentration (CMC) and subjecting them to four freeze–thawing cycles to induce bicelle formation prior to structural analysis (see the methods section for details). Figure 2 shows a summary of the hydrodynamic radii (R_H_) obtained from the DLS experiments and the radii of gyration (R_g_) obtained from the SAXS measurements of the samples. By exploring these phospholipids, we aimed to highlight the effects of the different natures of the tail and headgroup regions on the final assemblies obtained, a first step toward creating more representative models of mammalian and bacterial membranes. 

DLS measurements showed that most phospholipid/detergents mixtures investigated formed a monodisperse population of particles (Appendix A) with an increasingly large R_H_ that grew as a function of q. As q was inversely proportional to the molar fraction of detergent in the mixture, higher q values were expected to promote the formation of larger lipid aggregates, in line with previous results [17]. The particle R_H_ followed the trend DHPC > CHAPS > cholate for all phospholipid mixtures at nearly all q values investigated (except for the POPC/POPG-CHAPS mixtures at q 2.5 and 3, which exhibited a larger R_H_ than the DHPC mixtures). For SAXS-based R_g_, no estimation could be made for the conditions where large aggregates were present due to the lack of the Guinier regime. The q-dependent growth was considerably less marked for the sodium cholate-containing samples in comparison with DHPC and CHAPS for both saturated and unsaturated lipids. In all sodium cholate mixtures, the R_H_ values increased from a minimum of ~15 Å at q = 0.5 to a maximum of ~25 Å at q = 5. These values were too small to indicate the presence of a phospholipid bilayer structure; therefore, it was not likely that bicelles formed under these conditions, even at the highest q value tested (Figure 2, upper panels).

The DLS data did not provide any information regarding the internal structure of the observed particles besides the average R_H_. Therefore, it is not possible, based on these data alone, to determine what type of particles formed in the solution and whether they were bicelles or mixed micelles [18]. The R_H_ values and the q-dependent behavior of the particle sizes obtained from the DLS data suggest that, with sodium cholate, bicelles do not form in the presence of neither saturated nor unsaturated phospholipids. For CHAPS and DHPC, however, small discoidal bicelles could be representative of the structures formed with saturated lipids, whereas, in the presence of unsaturated lipids, DHPC and CHAPS might form small bicelles at low q values that quickly grow in size as the relative amount of detergent is reduced. The presence of negatively charged lipids in the mixtures had an overall minor effect on the R_H_.

SAXS measurements were performed to gain further insight into the structural characteristics of the particles studied by DLS. As for DLS, SAXS measurements were taken at 25 °C and the R_g_ values extracted from the SAXS curves are shown in Figure 2, lower panels. The R_g_ data obtained from the model-independent analysis of the SAXS data were in close agreement with the DLS-based R_H_ data, providing a similar picture of the q-dependent size changes in the particles.

The SAXS data were fitted to the elliptical bicelle model [17] implemented in SASview (http://www.sasview.org/, accessed on 1 October 2022) [19], consisting of a phospholipid bilayer modeled as an elliptical cylinder subdivided into three sections: heads, tails, and heads. The heads are symmetrical and are referred to as the disk face. The disk is surrounded by a belt or rim, representing the detergent’s hydrophilic moiety. The hydrophobic part of the detergent is assumed to be incorporated into the tail region of the disk. The SAXS data were fitted by fixing the scale factor to the calculated volume fraction of the components in the solution obtained by summing the contribution of the molecular volumes of each component adjusted by its concentration. The electron densities of the components were initially set to the calculated values (Appendix A) and allowed to vary to account for solvent penetration effects, while the thickness of the face, core, and rim were fitted together with the size of the cylinder’s major radius and the ratio between the major and minor radii. The best fits to the data (solid lines) are included in Figure 3, Figure 4A and Appendix A, and the parameters derived from such fits are shown in Figure 4B, Figure 5B and Appendix A, and the values are reported in Appendix A.

The bicelle model was found to produce adequate fits for all DMPC (Figure 3) and DMPC/DMPG (Appendix A) mixtures with the three detergents studied. The bilayer head groups (face) and core thickness values were comparable for bicelles composed of DHPC and CHAPS, and ranged around 23–25 Å for the tail region and 9–12 Å for the head groups, except for DMPC–CHAPS at q = 0.5. The headgroup and core thickness values are consistent with those of a DMPC lipid bilayer core in the fluid phase [20,21,22]. The rim thickness stabilized above q = 1.5 at approximately 8 and 4 Å for CHAPS and DHPC, respectively. The disks became more elongated with increasing q, growing by a factor of 1.6–2 from q = 0.5 to 5.

The data for the DMPC–sodium cholate mixtures clearly differed from those of the other DMPC–detergent mixtures. These particles were not only smaller in radius, but the bilayer core was significantly thinner compared with the particles formed with the other two detergents. The face and rim thickness were similar to those found for the DHPC and CHAPS mixtures. Thus, the overall shape of the DMPC–sodium cholate particle remained rather spherical, even at the highest q examined here, if we consider the total bilayer thickness (~10 × 2 + ~19 = ~49 Å) and the total disk diameter (~4 × 2 + ~25 × 2 = ~58 Å), together with the ellipticity value remaining close to 1. Therefore, it is plausible that DMPC–cholate mixtures form mixed micelles, rather than bicelles within this q range. The particle size and morphology for DMPC–detergent mixtures at the intermediate value of q = 2.5 was confirmed using cryo-EM (Appendix A). Small spherical particles were observed for sodium cholate and slightly elongated particles were seen for CHAPS and DHPC, with the largest particles formed by DMPC–DHPC mixtures, in agreement with the SAXS and DLS measurements.

The SAXS data and best fits for the mixtures composed of POPC and DHPC are shown in Figure 4. SAXS data for the other combinations of unsaturated phospholipids and detergents were also collected and are shown in the Appendix A; however, these measurements did not follow a clear q-dependent trend and could not be adequately fitted using the bicelle model employed for the other lipid–detergent mixtures. Our SAXS analysis rationale was based on the q-dependent evolution of the scattering curves, which shifted gradually at lower Q and lower intensities as the detergent concentration was lowered. This suggested a continuous evolution of the bicellar structure, which was reflected in the progressive increase in the radius (Figure 5). The absence of a clear trend in the unsaturated lipid data (except for the POPC–DHPC system) indicated that the change in the structure was of a more complex nature, and likely arose from the coexistence of non-homogeneous structures affected by the lipid:detergent ratio in a non-trivial manner. Such heterogeneous structural composition is no longer represented by the simple single-core-shell particle model or bicelle model. The only combination that resulted in a clear trend was the POPC–DHPC series, which was, therefore, modeled with the bicelle form factor. The parameters for the best fits obtained from the POPC–DHPC particles are shown in Figure 4 and Appendix A. The values obtained for the thickness of the head (face), core, and rim regions were comparable to those found for the corresponding saturated system. As indicated by the R_H_ and R_g_ values, however, a significant increase in the size took place as q increased beyond 1. This was reflected in an increase in the major radius of the particles, which reached ~950 Å at q = 5. 

The increase in the parameter describing the major radius was accompanied by a concomitant decrease in the ellipticity ratio. Notably, the resulting minor radius remained between 20 and 30 Å for all q values, suggesting that the structures grew almost exclusively in one direction into highly elongated particles that resembled rods more closely than large, flat disks, as indicated by the relatively constant values of the minor radius and bilayer thickness across the q ratios explored. With the increasing size, a Bragg peak at a Q value of 0.096 Å^−1^ became apparent at q = 2.5, which increased in intensity at q = 3 and at q = 5 (Figure 4C and Appendix A). Under the latter conditions, a faint second Bragg peak also started to appear at 0.192 Å^−1^, which was twice the Q value of the first one. The primary Bragg peak corresponded to repeating structures with a correlation distance of ~65 Å. Given the elongated nature of the structures, it is reasonable to assume that this distance corresponded to the distance between the centers of two adjacent, side-by-side rods, resulting in a separation of 15–20 Å between their hydrophilic external regions. Bragg peaks did not appear in any other lipid detergent combination measured here and were also absent from the POPC–POPG DHPC sample, indicating that the presence of 20% POPG in the mixture was already sufficient to change the internal organization.

Finally, an attempt was made to produce bicelles using natural phospholipids from polar extracts of *E. coli* (Figure 6). DLS data were collected for mixtures with DHPC, cholate, and CHAPS. Small particles were formed for DHPC at q = 0.5 (R_H_ < 10 nm), while a broad particle size distribution was found for larger q values, ranging within 15 nm < R_H_ < 100 nm. Mixed micelles and vesicles or worm-like micelles were most likely formed at low and high q values, respectively. Additional studies with detailed DLS measurements and EM image acquisition would be necessary to identify the exact structures formed, but this is beyond the scope of the present work. For mixtures with sodium cholate, on the other hand, small particles with R_H_ < 10 nm were found for all q values. This suggests that small mixed micelles were formed in most cases, as was the case for the other lipids studied. Finally, a broad size distribution was found for mixtures with CHAPS ranging from a few nanometers in size (for q < 1.5) to approximately 100 nm for q > 2. This suggests that either small mixed micelles or vesicles/worm-like micelles were formed in this case as well.

## 3. Discussion

The saturated phospholipid mixtures with DHPC allowed bicelles in the q range studied to be obtained at 25 °C in TBS, except for q = 0.5 (Figure 2, Figure 3, Figure 4 and Figure 5). Bicelles were also obtained for saturated phospholipid mixtures with CHAPS. This was established by the SAXS measurements (Figure 5), which demonstrated a clear bilayer structure with symmetric heads and a lipid core having layer thicknesses consistent with the values expected for fluid, DMPC bilayers [20,22] when using saturated phospholipid mixtures with CHAPS or DHPC. For q = 0.5, the core thickness was lower than 20 Å and the disk radii were 30 and 20 Å for DHPC and CHAPS, respectively. This suggests that mixed micelles, rather than disks, were formed in this case. This is in agreement with a previous study of DMPC–DHPC mixtures at q < 1 [18]. CHAPS is a detergent often used in membrane protein purification [23] as it maintains the functionality of the protein of interest. Earlier NMR studies also found strong indications that bicelles could successfully be formed by using DMPC and CHAPS [14]. Therefore, CHAPS can, with confidence, be used instead of DHPC to form bicelles containing membrane proteins in the q-range of 1 < q < 5. Indeed, it was previously found that the use of CHAPS increased the stability of the transmembrane protein Opsin in DMPC–detergent bicelles [24]: this protein is particularly unstable in detergent or in DHPC-based bicelles.

Sodium cholate is another detergent often used in membrane purification. Mixtures of saturated phospholipids with sodium cholate exhibited no significant dependency of the lipid–detergent q range on particle size (Figure 2). Moreover, the lipid core was thinner than expected for a fluid DMPC bilayer (Figure 5). This suggests that mixed micelles are formed, rather than bicelles, when using sodium cholate as a detergent. Accordingly, cryo-EM corroborated the lack of a disk-like appearance for DMPC–cholate mixtures as compared with DMPC–DHPC or CHAPS mixtures at q = 2.5 (Appendix A).

For unsaturated phospholipids, bicelles were only found under very limited experimental conditions (for q = 0.5 and 1). Higher q values produced particles with significantly larger sizes. This is expected, as DMPC–DHPC bicelles tend to merge and grow into vesicles at temperatures well above the gel–fluid phase-transition temperature, T_m_, for DMPC [25] or upon further increasing q [26]. The T_m_ of POPC and POPG is around 0 °C. Therefore, these lipids are in the fluid phase at room temperature, and it is expected that elongated disks or vesicles are formed at intermediate q values for both DHPC and CHAPS. Indeed, such structures have been observed previously for DMPC–DHPC at q = 3–4 measured in 20 mM HEPES buffer at pH 7.4 enriched with 150 mM NaCl [27]. Detailed microscopy studies would enable whether elongated bicelles or vesicles are formed at intermediate and high q values for unsaturated phospholipids to be determined. Further determination of the non-bicellar structures formed by the mixtures hereby presented beyond the scope of the present work.

Polar *E. coli* extracts have previously been shown to produce bicelles at q = 0.5 when mixed with DHPC, as studied by NMR, yielding R_H_ = 39 ± 5 Å [28]. The authors suggested that the larger size (and variation) found in R_H_ was due to the longer lipid tails present in the *E. coli* bicelles as compared with the traditional DMPC/DHPC bicelles. Indeed, the authors found that the composition of the bicelles depended on whether the lipid extracts were taken from cells grown at 20 or 37 °C, with major differences found in the main lipid types present, as well as the acyl length. This could be related to the natural extract mixture being above or below their respective T_m_, although this transition is rarely sharp in natural lipid extracts [29]. For POPC/POPG–DHPC at q < 1, R_H_ is ~40 Å (Figure 2), while, for the polar *E. coli* extracts, R_H_ is ~8 and 50 Å for q = 0.5 and 1, respectively (Figure 6). Thus, our data suggest that bicelles produced with polar *E. coli* lipid extracts could potentially only be obtained at q = 1 for DHPC. However, the polar *E. coli* extract–DHPC mixtures exhibited broad size distributions with no clear trend with q, as measured by DLS (Figure 6); thus, it is not certain whether the structures observed correspond to bicelles at all. The use of cholate or CHAPS as detergents with polar *E. coli* lipid extracts did not show any promising results regarding the formation of bicelles.

## 4. Materials and Methods

### 4.1. Materials

The lipids phospholipid 1,2-dimyristoyl-sn-glycero-3-phosphocholine (DMPC), 1,2-dimyristoyl-sn-glycero-3-phospho-sn-glycerol (DMPG), 2-oleoyl-1-palmlitoyl-sn-glyecro-3-phosphocholine (POPC), and 1-palmitoyl-2-oleoyl-sn-glycero-3-phospho-(1′-rac-glycerol) (POPG); *E. coli* polar extract; and the detergents 1,2-dihexanoyl-sn-glycero-3-phosphocholine (DHPC) and CHAPS were purchased from Avanti Polar Lipids Inc., Alabaster, AL, USA. Sodium cholate was purchased from Sigma Aldrich, St. Louis, MO, USA.

### 4.2. Bicelles Preparation

For the preparation of the lipids and detergent stock solutions in organic solvents, phospholipids and DHPC were dissolved in chloroform, sodium cholate in methanol, and CHAPS in a 4:1 methanol:chloroform mixture. The latter required sonication and heating to 40 °C to completely dissolve. Lipid films were prepared by mixing phospholipids and detergents at the desired q to obtain a final phospholipid concentration of 5 mg/mL in all samples. Lipids and detergents were mixed in organic solvents at seven different lipid:detergent molar ratios to obtain q values of 0.5, 1, 1.5, 2, 2.5, 3, and 5, and the samples were dried overnight using a miVac Quattro concentrator (Genevac Ltd., Ipswich, UK). The dry lipid:detergent films were resuspended at a lipid concentration of 5 mg/mL in TBS buffer at pH 7.4 containing the desired detergent at a concentration just above the nominal critical micelle concentration (CMC). The detergent concentrations in the buffers used were 5 mM, 10 mM, and 16 mM for CHAPS, DHPC, and sodium cholate, respectively. The samples were subjected to three cycles of freeze–thawing using liquid nitrogen and a water bath heated to 45 °C, and vortexed for 1 min after each cycle. Samples were then centrifuged at 13,000 rpm for 10 min and the supernatants were used for further analysis.

### 4.3. Dynamic Light Scattering (DLS)

The size of the different bicelle preparations was determined using multi-angle dynamic light scattering (MADLS) on a Zetasizer Ultra (Malvern Panalytical, Ltd., Malvern, UK) using a quartz cuvette (ZEN2112) with a final volume of 90 μL. All measurements were performed on freshly prepared bicelle samples, mostly in duplicate or triplicate, at 25 °C and data were processed using the software Zetasizer Ultra-Pro ZS Xplorer v1.31.

### 4.4. Small-Angle X-ray Scattering (SAXS)

SAXS measurements were performed at beamline P12 operated by EMBL Hamburg at the PETRA III storage ring (DESY, Hamburg, Germany). Measurements were performed using an X-ray beam of 10 kEV, corresponding to a wavelength of 1.24 Å, a beam size of 250 × 100 μm^2^ (FWHM), and a flux of 5.10^12^ photons per second.

Samples and buffers were loaded in a flow-through capillary using the beamline sample changer and flown during X-ray exposure to limit radiation damage. Scattered photons were collected on a Pilatus 6M detector (Dectris, Baden, Switzerland). With a sample-to-detector distance of 3 m, the SAXS intensities can be collected for Q ranging between 0.0027 Å^−1^ and 0.74 Å^−1^.

Data were also collected on the buffer solution used to resuspend the bicelles. These data contained the scattering of the buffer and of the instrumental background that can be subtracted from the sample curve to obtain the bicelle form factor.

Initial data reduction was performed using the data analysis pipeline SASFLOW [30]. For each measurement, 40 successive frames with exposure of 100 ms were collected. The images obtained were radially average. The successive frames were compared, and those affected by radiation damage were discarded. The remaining frames were averaged, and the corresponding buffer curve was subtracted to obtain the form factor of the micelles.

Data obtained using the laboratory SAXS instrument (XEUSS3.0, Xenocs, Grenoble, France) at Malmö University showed consistent results on replicate samples, albeit with a slightly reduced Q range and with reduced counting statistics. The agreement between the synchrotron and laboratory SAXS supports the stability and reproducibility of the bicelle preparation, and the absence of any effects of radiation damage.

### 4.5. SAXS Analysis

Calculations of the radius of gyration (R_g_) were performed using the AUTORG function [31] available in the Primus software version 3.0, EMBL, Germany [32].

SAXS data were fitted using SasView 4.2.2 [19]. Bicelles were modeled with the form factor of a core–shell elliptical cylinder described in the software documentation [33]. Briefly, the hydrophobic core of the bicelles was modeled as a cylinder with an elliptical base defined by a major and minor radius. The hydrophilic lipid head groups were described by two symmetric cylinders adjacent to the bottom and top faces of the central cylinder, while the detergent ring surrounded it, covering the length of the hydrophobic core, as described in Figure 5. The scale factor that defined the intensity of the simulated data was set to match the volume fraction of scattering components calculated by adding together the molecular volumes of the species in solution according to their concentrations (Appendix A) and allowed to vary within a 10% range from its calculated value. The same molecular volumes were used to estimate the electron densities of the components to define the starting values of the fits. The electron density of the hydrophobic core was kept fixed to its calculated value, while that of the head group was allowed to vary to consider solvent penetration within the hydrophilic regions. 

### 4.6. Cryogenic Electron Microscopy (Cryo-EM)

Cryo-EM image acquisition was performed on a JEM-2200FS transmission electron microscope (JEOL, Tokyo, Japan) at the National Center for High-Resolution Electron Microscopy (nCHREM) at Lund University. The microscope was equipped with a field-emission electron source, a cryo pole piece in the objective lens, and an in-column energy filter (omega filter). Zero-loss images were recorded at an acceleration voltage of 200 kV on a TemCam-F416 camera (TVIPS) using SerialEM under low-dose conditions. 

Samples were prepared using an automatic plunge freezer system (Leica EM GP, Wetzlar, Germany) with the environmental chamber set to 21 °C and 90% relative humidity. A 4 µL droplet of the sample solution was deposited on a lacey formvar carbon-coated grid (Ted Pella, Redding, CA, USA) and was blotted with filter paper to remove excess fluid. The grid was then plunged into liquid ethane (around −183 °C) to ensure rapid vitrification of the sample in its native state. The specimens were then stored in liquid nitrogen (−196 °C) and, prior to imaging, transferred to the microscope using a cryo transfer tomography holder (Fischione, Export, PA, USA, Model 2550).

## 5. Conclusions

This study clearly shows that bicelles can be obtained with saturated phospholipids (DMPC and DMPG) using both DHPC and CHAPS as detergents for q > 1. Mixed micelles were likely formed for q < 1. The study further demonstrates that no bicelles can be obtained when using sodium cholate as a detergent, regardless of the q used, and instead very small mixed micelles are formed. In the case of unsaturated phospholipids (POPC and POPG), bicelles can be obtained by using both DHPC and CHAPS as detergents, but only for q < 1

Finally, our results show that bicelles cannot be successfully produced with *E. coli* polar extracts, at least with the detergents used in this study and within the q range used. Note that the experimental conditions in this study were saline TBS buffer at pH 7.4 and 25 °C, which are physiologically relevant conditions suitable for the study of transmembrane and membrane-bound proteins.

## Figures and Tables

**Figure 1 molecules-27-07628-f001:**
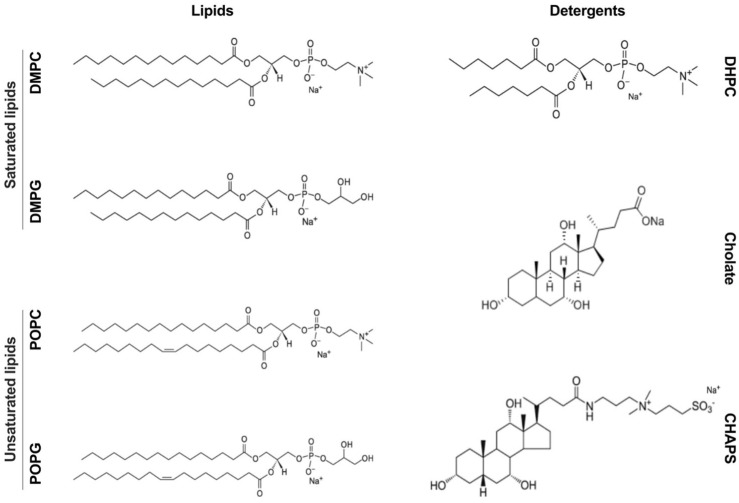
Schematic overview of the phospholipids and detergents used to explore bicelle formation in this study.

**Figure 2 molecules-27-07628-f002:**
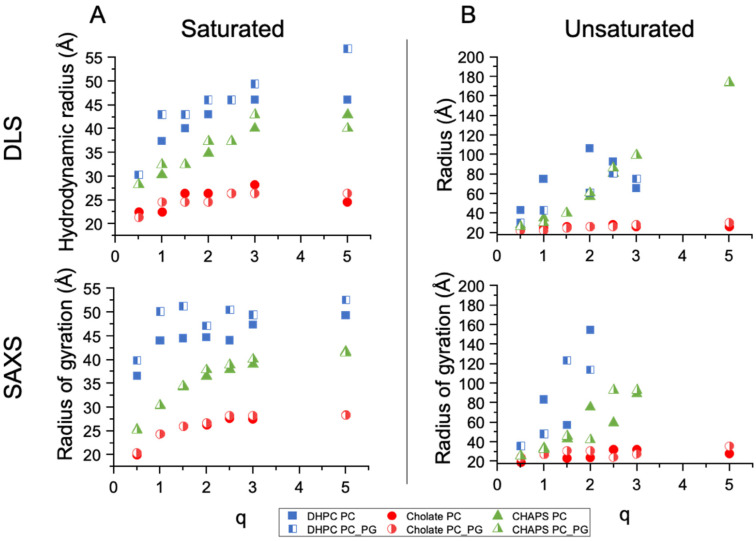
DLS (top) and SAXS (bottom)-derived radii of bicelles for (**A**) DMPC and DMPC:DMPG 80:20 and (**B**) POPC and POPC:POPG 80:20 mol%. Data were measured at 25 °C in tris-buffered saline (TBS). The radius of gyration could not be estimated for unsaturated lipids mixed with CHAPS at q = 5 and unsaturated lipids mixed with DHPC at q values of 2.5, 3, and 5 due to the absence of the Guinier regime. The DLS curves are representative of one to three independent experiments. Most of the SAXS data were measured in duplicate.

**Figure 3 molecules-27-07628-f003:**
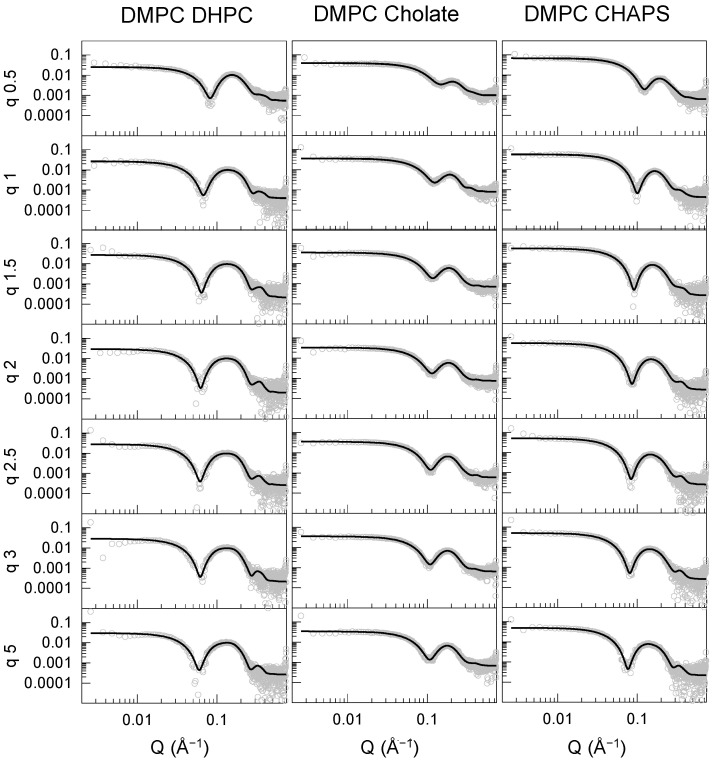
SAXS profiles (data points) and best fit to the bicelle model (lines) for DMPC particles composed of DHPC (**left**), cholate (**center**), and CHAPS (**right**) at seven different q ratios measured at 25 °C in TBS. Most of the SAXS data were measured in duplicate.

**Figure 4 molecules-27-07628-f004:**
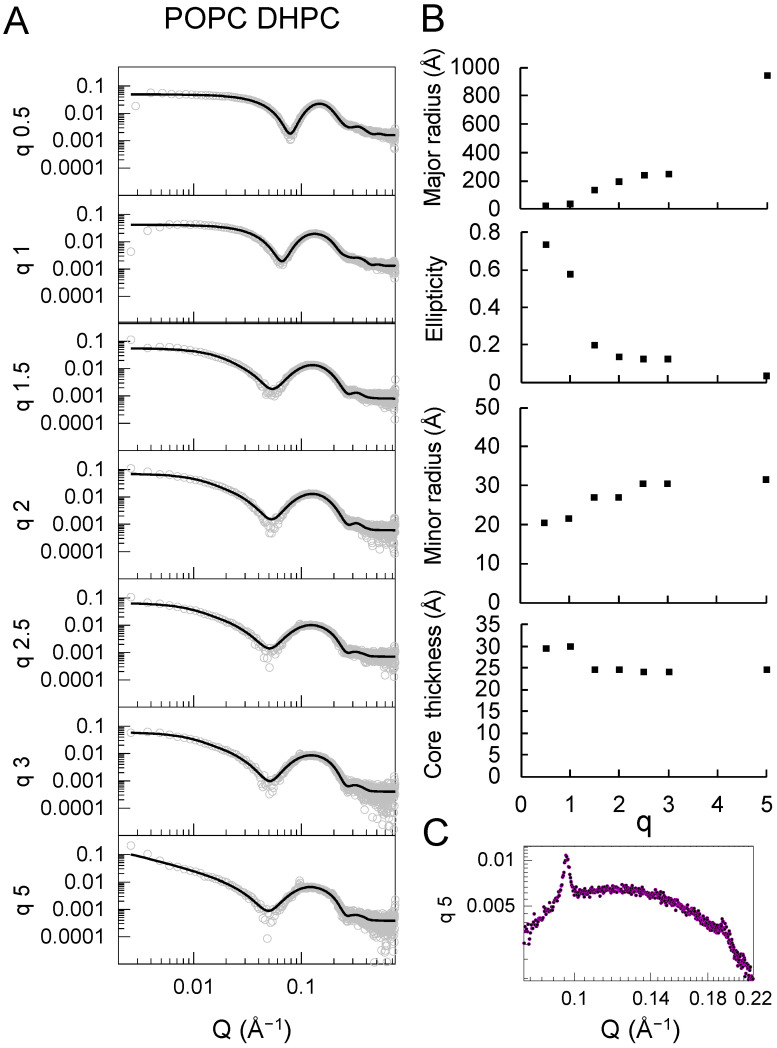
(**A**) SAXS curves (data points) and fit lines for bicelles composed of POPC and DHPC at 7 different q ratios. (**B**) Parameters derived from the SAXS fits. (**C**) Bragg peak in the SAXS curve for the POPC:DHPC mixture at q = 5.

**Figure 5 molecules-27-07628-f005:**
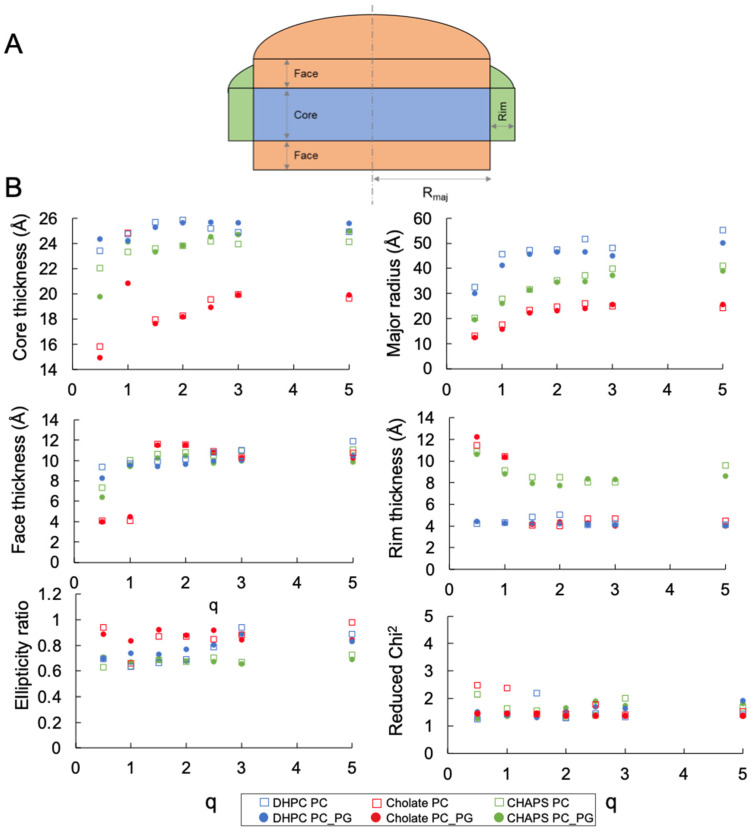
(**A**) Cartoon representing the bicelle model implemented in SASview used to fit the SAXS data. The ellipticity ratio defines the minor radius orthogonal to the major radius in the bilayer plane (**B**) Parameters of the best bicelle fits for the SAXS curves shown in Figure 3. Same colour scheme as in Figure 2. The full list of values is given in Appendix A.

**Figure 6 molecules-27-07628-f006:**
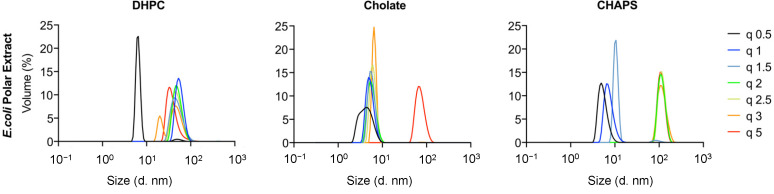
DLS size distribution of mixtures of DHPC, cholate, and CHAPS with an *E. coli* polar lipids extract. The curves are representative of one to three independent experiments.

## Data Availability

All SAXS curves shown in the main manuscript and Appendix A have been deposited in the SASBDB database and are available for download at https://sasbdb.org (accessed on 1 October 2022). SASBDB entries for the datasets are: SASDQL7—DMPC DHPC—https://www.sasbdb.org/data/SASDQL7 (accessed on 1 October 2022); SASDQM7—DMPC cholate—https://www.sasbdb.org/data/SASDQM7 (accessed on 1 October 2022); SASDQN7—DMPC CHAPS—https://www.sasbdb.org/data/SASDQN7 (accessed on 1 October 2022); SASDQP7—DMPC DMPG and CHAPS—https://www.sasbdb.org/data/SASDQP7 (accessed on 1 October 2022); SASDQQ7—DMPC DMPG and cholate—https://www.sasbdb.org/data/SASDQQ7 (accessed on 1 October 2022); SASDQR7—DMPC DMPG and DHPC—https://www.sasbdb.org/data/SASDQR7 (accessed on 1 October 2022); SASDQS7—POPC CHAPS—https://www.sasbdb.org/data/SASDQS7 (accessed on 1 October 2022); SASDQT7—POPC cholate—https://www.sasbdb.org/data/SASDQT7 (accessed on 1 October 2022); SASDQU7—POPC DHPC—https://www.sasbdb.org/data/SASDQU7 (accessed on 1 October 2022); SASDQV7—POPC POPG and CHAPS—https://www.sasbdb.org/data/SASDQV7 (accessed on 1 October 2022); SASDQW7—POPC POPG and cholate—https://www.sasbdb.org/data/SASDQW7 (accessed on 1 October 2022); SASDQX7—POPC POPG and DHPC—https://www.sasbdb.org/data/SASDQX7 (accessed on 1 October 2022).

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
