# Peer review of "Expanding the Toolbox for Bicelle-Forming Surfactant–Lipid Mixtures"

_molecules, 2022, doi:10.3390/molecules27217628_

Round 1
Reviewer 1 Report
The authors present analysis of bicelles in different lipid compositions using SAXS, DLS and some CryoEM examples in the supplementary materials. They expand the fundamental characterisation data of lipid and detergent mixtures toolkits. The fits to the DMPC data with DHPC, cholate and CHAPS are nice and look clear in the publication. It would be nice to include the fit parameters in tables in the supplementary data to enable other researchers to use the data as a resource when searching for bicelles with specific properties. It is a shame that the SAXS data for the POPC, POPC-POPG with CHAPS, POPC,POPC-POPG data with sodium cholate and the POPC-POPG data are included as raw data and without fits. The authors mention that the bicelle model did not give adequate fits – have they tried other models / CryoEM to look at how the structures might be different? For the DLS data it would also be nice to clarify ‘N’ (I assume 1?) and if triplicate measurements were performed on each individual sample. Furthermore, expanding the CryoEM and using the images to validate the bicelle sizes obtained would be a nice addition but depends on access to instrumentation and is not essential. In my opinion understanding the different non bicellar structures and characterising them would be an interesting avenue and could be used to create phase diagrams of the structures which could be used as a tool for bicelle choice and experimental design but appreciate this might be out of the scope for the study. Otherwise the study is well written, with clear figures and only minor typos in the text.
Author Response
We are very grateful to the two reviewers for carefully reading our manuscript and for providing comments and suggestions that certainly helped us to further improve the quality of our work. We have now addressed the reviewers’ comments and suggestions in the point-by-point response below.
All the corrections/changes are evidenced in track change along the revised version of the submitted manuscript.

Reviewer 2 Report
The manuscript entitled “Expanding The Toolbox for Bicelle Forming Surfactant-Lipid Mixtures” could impact the science community. Characterization techniques dynamic light scattering, small-angle X-ray scattering, and cryogenic electron microscopy are appropriately used to explain sample characterization. The manuscript is well organized, and all experiments are conducted following the standards, but some issues should be corrected before publication.
How did the Authors decide on the combination of these bicelles? Is this a rational criterion, or did the authors followed a specific model, or is the choice based just on your experience?
Please evaluate the solubility of the bicelles in water (including acid, neutral, alkaline solution, and pure water).
Minor remarks
Line 174
Written: Figure S2
Comment: Authors did not provide supplementary material
Line 150
Written: Figure S3
Comment: Authors did not provide supplementary material
Line 183-184
Written: supplementary material (Figure 183 S4-S6)
Comment: Authors did not provide supplementary material
Line 187
Written: Supporting Information Figure S7
Comment: Authors did not provide supplementary material
Line 199
Written: Supporting Information Figure S8
Comment: Authors did not provide supplementary material
Author Response

(The authors gave the same response as above.)
